# Astrocytes Undergo Metabolic Reprogramming in the Multiple Sclerosis Animal Model

**DOI:** 10.3390/cells12202484

**Published:** 2023-10-19

**Authors:** Sofia Pereira das Neves, João Carlos Sousa, Ricardo Magalhães, Fuying Gao, Giovanni Coppola, Sebatien Mériaux, Fawzi Boumezbeur, Nuno Sousa, João José Cerqueira, Fernanda Marques

**Affiliations:** 1Life and Health Sciences Research Institute (ICVS), School of Medicine, Campus Gualtar, University of Minho, 4710-057 Braga, Portugal; sofianeves4@gmail.com (S.P.d.N.); jcsousa@med.uminho.pt (J.C.S.); njcsousa@med.uminho.pt (N.S.); jcerqueira@med.uminho.pt (J.J.C.); 2ICVS/3B’s PT Government Associate Laboratory, 4806-909 Guimarães, Portugal; 3NeuroSpin, CEA, Paris-Saclay University, Centre d’études de Saclay, Bâtiment 145, 91191 Gif-sur-Yvette, Francesebastien.meriaux@cea.fr (S.M.); fawzi.boumezbeur@cea.fr (F.B.); 4Program in Neurogenetics, Department of Neurology, David Geffen School of Medicine, University of California, Los Angeles, CA 90095, USA; fygao@ucla.edu (F.G.); gcoppola@ucla.edu (G.C.); 5Clinical Academic Center, 4710-243 Braga, Portugal

**Keywords:** multiple sclerosis, astrocytes, metabolism, glycolysis, TCA cycle

## Abstract

Multiple sclerosis (MS) is a chronic inflammatory disease of the central nervous system that presents a largely unknown etiopathology. The presence of reactive astrocytes in MS lesions has been described for a long time; however, the role that these cells play in the pathophysiology of MS is still not fully understood. Recently, we used an MS animal model to perform high-throughput sequencing of astrocytes’ transcriptome during disease progression. Our data show that astrocytes isolated from the cerebellum (a brain region typically affected in MS) showed a strong alteration in the genes that encode for proteins related to several metabolic pathways. Specifically, we found a significant increase in glycogen degradation, glycolytic, and TCA cycle enzymes. Together with these alterations, we detected an upregulation of genes that characterize “astrocyte reactivity”. Additionally, at each disease time point we also reconstructed the morphology of cerebellum astrocytes in non-induced controls and in EAE animals, near lesion regions and in the normal-appearing white mater (NAWM). We found that near lesions, astrocytes presented increased length and complexity compared to control astrocytes, while no significant alterations were observed in the NAWM. How these metabolic alterations are linked with disease progression is yet to be uncovered. Herein, we bring to the literature the hypothesis of performing metabolic reprogramming as a novel therapeutic approach in MS.

## 1. Introduction

Multiple sclerosis (MS) is a chronic progressive inflammatory disease of the central nervous system (CNS). It is a complex disease, affecting 2.8 million people worldwide, in which there is an immune response against the myelin sheath of CNS axons, but its underlying mechanisms are only partially understood.

In terms of disease mechanisms, it is known that the patient’s immune system starts to attack its own CNS. Once the peripheral immune cells reach the CNS, they attack a neuronal structure called myelin. This attack is further potentiated by glial cells that reside in CNS tissues. This neuronal demyelination leads ultimately to neurodegeneration, resulting in the permanent loss of neurological functions including walking and bladder control [1]. Today, MS treatments are focused on drugs that work by suppressing the immune system [2]. While not curing MS, they slow down disease activity while reducing the severity and frequency of flare-ups. In addition to their high costs, most of the treatments have an FDA indication for the relapse–remitting disease forms but not for the progressive forms. Therefore, our mission is to lead global MS efforts and improve patients’ life quality for the progressive forms as well. The reason for this unsatisfactory situation is that the disease mechanisms driving progressive MS remain unresolved. However, one intriguing observation is the fact that, in patients in the progressive stage of the disease, local glial cells are also relevant and able to potentiate inflammation alone, leading to neurodegeneration by themselves which suggests that glial cells are key in MS [3].

Among glial cells mediating local inflammatory responses in the brain (astrocytes and microglia), astrocytes seem to be key regulators upon CNS injury [4]. Depending on timing and context, reactive astrocytes may lose homeostatic functions and gain other phenotypes and functions. Whether the overall impact on disease is beneficial or detrimental will be determined by the balance and nature of lost and gained functions and the relative abundance of different astrocyte subpopulations [5,6]. Thus, in order to develop new strategies for MS treatment, it is crucial to understand the glia response during MS progression. In line with this hypothesis, we have recently used an animal model of MS to perform high-throughput sequencing of cerebellar astrocytes and determine their transcriptome. The study of cerebellar astrocytes in MS is relevant since cerebellar abnormalities are continuously demonstrated to be associated with a variety of motor or non-motor dysfunctions. Also, we have previously showed that, specifically, cerebellar astrocytes are activated in the experimental autoimmune encephalomyelitis (EAE) MS animal model and that their activation is associated with the EAE motor problems [7]. Overall, we found an upregulation of regulatory genes of the glycolytic and tricarboxylic acid (TCA) cycle pathways at the onset of the disease. However, how these metabolic alterations are linked with disease progression and with a new possible MS treatment, by performing metabolic reprograming, is yet to be uncovered.

## 2. Materials and Methods

### 2.1. Animals and EAE Induction

All experiments were reviewed and approved by the Portuguese national authority for animal experimentation, Direcção Geral de Veterinária (ID: DGV9458). Animals were housed and handled in accordance with the guidelines for the care and handling of laboratory animals in the Directive 2010/63/EU of the European Parliament and Council.

Animals were housed under specific-pathogen-free conditions and maintained under standard laboratory settings: 12 h light/dark cycles (lights on at 8 a.m.), relative humidity of 55%, temperature between 22 and 24 °C, and fed with regular rodent chow (4RF21, Mucedola SRL, Milanese, Italy) and tap water ad libitum.

Disease was induced in 10-week-old female C57BL/6J mice purchased from Charles River Laboratories (France) using a commercial kit (EK-2110; Hooke Laboratories, Lawrence, MA, USA) according to the manufacturer’s instructions. As previously shown, C57BL/6 mice induced with MOG35-55 present a chronic course of disease [8,9] characterized by ascending paralysis resulting from the preferential attack to the spinal cord [10]. The disease is characterized in the beginning by a limp tail, which progresses to hind and forelimbs paralysis [11]. Also, multifocal and confluent areas of mononuclear inflammatory infiltrates and perivascular inflammatory cuffing in the cerebellum and hindbrain white matter are observed. The chronic disease course makes this a good model to study SP-MS [12]. The MOG35-55-induced C57BL/6 model is particularly important in MS studies due to the increased availability of gene-modified strains on this background [13,14]. Herein, we used females for the immunization, because females have more spinal cord infiltrating cells and demyelination than males in spite of an essentially identical EAE disease severity [15]. Briefly, animals were immunized subcutaneously with 200 μg of myelin oligodendrocyte glycoprotein (MOG)_35−55_, emulsified in complete Freund’s adjuvant (CFA), at the upper and lower back. Pertussis toxin (PTX) in phosphate-buffered saline (PBS) was administered intraperitoneally 2 and 24 h after immunization [136 ng of PTX per injection (lot #1006)]. Non-induced age-matched littermate females were used as controls and were injected subcutaneously with a control emulsion and PTX (CK-2110; Hooke Laboratories) at the same concentration and time points as the EAE animals. Animals were weighed daily and monitored for clinical symptoms of disease.

Disease severity was assessed daily as follows: 0 = no clinical symptoms; 0.5 = partially limp tail; 1 = paralyzed tail; 1.5 = at least one hind limb falls through consistently when the animal is placed on a wire rack; 2 = loss in coordinated movement, wobbly walk; 2.5 = dragging of hind limbs; 3 = paralysis of both hind limbs; 3.5 = hind limbs paralyzed and weakness of forelimbs; 4 = complete hind limbs paralysis and partial forelimbs paralysis; 4.5 = animal is not alert, no movement; 5 = moribund state or death. Paralyzed mice, with clinical scores above 3, were offered easier access to food and water.

For biological sample collection, groups of EAE and non-induced animals were sacrificed, at the light phase of the diurnal cycle, at Day 6 post-immunization (p.i.) (pre-symptomatic phase), on the first day of a clinical score of 3 (onset/peak phase; Days 9–12), and on Days 20–21 p.i. (chronic phase). Animals were anesthetized with an intraperitoneal injection of ketamine hydrochloride (150 mg/kg, Imalgene 1000) plus medetomidine hydrochloride (0.3 mg/kg, Dorben). Under deep anesthesia, mice were transcardially perfused with cold 0.9% saline solution, and the brain was dissected. For histological analysis, the brain was immediately embedded in the Tissue-Tek O.C.T. compound (Sakura Finetek, Japan), snap-frozen, and kept frozen (−20 °C) until further sectioning.

### 2.2. Astrocyte Isolation

The cerebellum was macrodissected for astrocyte isolation using magnetic-activated cell sorting (MACS) [16]. Briefly, the cerebellum was mechanically dissociated using a scalpel, followed by enzymatic dissociation using the Neural Tissue Dissociation kit (P) (Miltenyi Biotec, Cologne, Germany), according to the manufacturer’s instructions. Myelin and cell debris were removed using the Myelin removal kit (Miltenyi Biotec), followed by microglia removal using Cd11b microbeads (Miltenyi Biotec). Finally, astrocytes were isolated using anti-astrocyte cell surface antigen (ACSA)-2 beads (Miltenyi Biotec). All the separation steps were performed using the AutoMACS Pro Separator equipment (Miltenyi Biotec).

### 2.3. Gene Expression Analysis by RNA-Sequencing

Total RNA was extracted from isolated astrocytes using the RNeasy Plus Micro kit (Qiagen, Hilden, Germany) according to the manufacturer’s instructions. RNA quality and quantification were performed using the Experion RNA HighSens Analysis kit (Bio-Rad, CA, USA) according to the manufacturer’s instructions.

RNA samples were sequenced at the UCLA Neuroscience Genomics Core. RNA sequencing (RNAseq) was carried out using Nugen Ovation RNA Ultra Low Input and Kapa Hyper. The RNA samples were made into a barcoded cDNA library and then sequenced at 2 × 75 bp paired end reads output with Illumina HiSeq 4000. No read trimming or filtering was performed with this dataset, because the quality distribution and variance appeared normal. Short reads were aligned using STAR to the mouse (mm10), with default parameters. Differential expression analysis was performed using an observation based-model (limma-voom). Genes with counts per million (CPM) > 0.5 in at least 3 non-induced samples and adjusted *p*-values < 0.05 were considered statistically significant and were used for differential expression analysis (Appendix A). Pathway analysis was performed using the ConsensusPathDB-mouse tool [17,18]. A total of three non-induced and four EAE animals were sacrificed per experimental time point for the RNAseq experiments.

### 2.4. Gene Expression Analysis by qRT-PCR

Total RNA was extracted from isolated astrocytes using the Rneasy Plus Micro kit (Qiagen, Hilden, Germany) according to the manufacturer’s instructions. RNA quality assessment and quantification were performed using the Experion RNA HighSens Analysis kit (Bio-Rad, Hercules, CA, USA) according to the manufacturer’s instructions. In total, 750 pg of total RNA from each sample was reverse transcribed into cDNA using the iScript cDNA synthesis kit (Bio-Rad), according to the manufacturer’s instructions. qRT-PCR was performed on a CFX96 real-time instrument (Bio-Rad) using the SsoFast EvaGreen Supermix (Bio-Rad). For each reaction, 5 µL of reaction mix, 0.5 µL of each primer (initial concentration 10 µM), 3 µL of Rnase/Dnase free water, and 1 µL of cDNA were used. The cycling parameters were 1 cycle at 95 °C, for 1 min (min), followed by 40 cycles at 95 °C for 15 s (s), annealing temperature (primer-specific) for 20 s and 72 °C for 20 s, finishing with 1 cycle at 65 °C to 95 °C for 5 s (melting curve). Product fluorescence was detected at the end of the elongation cycle. All melting curves exhibited a single sharp peak at the expected temperature. Adenosine Triphosphate subunit 5 beta (Atp5b), Heat Shock Protein 90 alpha family class B member 1 (Hspcb), and TATA binding protein (Tbp) were used as reference genes. Primers used to measure the expression levels of selected mRNA transcripts by qRT-PCR were designed using the Primer-BLAST tool of NCBI (Bethesda, MD, USA) on the basis of the respective GenBank accession numbers, or were used as described by Liddelow and colleagues (2017) [19]. Primers DNA sequences and annealing temperatures are provided in Appendix A. A total of three animals were used in the non-induced pre-symptomatic group; four animals in the non-induced onset and EAE pre-symptomatic groups; five animals in the EAE chronic group; and six animals in the non-induced chronic and EAE onset groups.

### 2.5. GFAP Immunofluorescence and 3-Dimensional Reconstruction of Astrocytes

Serial 20 µm sections of cerebellum were fixed in 4% paraformaldehyde in PBS for 30 min at room temperature (RT). After antigen retrieval, with pre-heated citrate buffer (Sigma-Aldrich, St. Louis, MO, USA) in the microwave for 20 min, tissue slices were permeabilized with PBS-triton 0.3% at RT for 10 min, and subsequently blocked with 10% fetal bovine serum in PBS-triton 0.3% at RT for 30 min. Slides were incubated overnight with rabbit anti-mouse GFAP antibody (1:200; Dako, Glostrup, Denmark), diluted in blocking solution. Afterwards, slides were incubated with Alexa Fluor^®^ 594 donkey anti-rabbit (1:500; Fisher Technologies, Thermo Fisher Scientific, Waltham, MA, USA), diluted in PBS-triton 0.3%, for 2 h at RT. After incubation with 4′,6-diamidino-2-phenylindole (DAPI; 1:200; Invitrogen, Thermo Fisher Scientific), for 10 min at RT, slides were cover-slipped with Immumount (Fisher Scientific, Thermo Fisher Scientific) and examined under fluorescent light.

To perform the 3-dimensional reconstruction of astrocytes, 3–4 photographs per animal were taken from the cerebellum white matter, using a confocal microscope (FV1000, Olympus) and the following parameters: 40× objective, 1024 × 1024 resolution, 1 µm increment. In the case of EAE animals, photographs were acquired both near lesion regions and in regions of normal-appearing white matter (NAWM), except for animals at the pre-symptomatic phase which did not present lesions. The confocal images were then used to performed the morphological reconstruction using the Fiji plugin “Simple Neurite Tracer” [20,21], as previously described [22]. For Sholl analysis, concentric circles were superimposed on astrocytes, with the origin in the cell soma and with a 4 µm distance from each other. The results are presented as the average of 4–5 animals per experimental group (n = 4 for non-induced groups and n = 5 for EAE groups), and 7–8 astrocytes were reconstructed per animal. The average of all astrocytes per animal was used for statistical analysis.

### 2.6. Statistical Analysis

Statistical analysis was performed using SPSS software (version 23, IBM, Armonk, NY, USA) and GraphPad Prism (version 8, La Jolla, CA, USA). The number of biological replicates (n) are specified in the legend of each figure. The results are presented as the mean ± standard error of the mean (SEM), or only as the mean for astrocytic Sholl analysis. Statistical significance was considered for *p* < 0.05 (*), *p* < 0.01 (**), *p* < 0.001 (***), *p* < 0.0001 (****). The partial eta squared value (η_p_^2^) was calculated as a measure of effect size [23].

## 3. Results

### 3.1. Altered Astrocytic Gene Expression in EAE Animals

To explore the alterations occurring in astrocytes throughout disease development, we performed transcriptomic analysis, by RNAseq, in astrocytes isolated from the cerebellum at different disease time points (Figure 1A). The cerebellum is known to be affected both in the human disease and in the EAE model, and astrocyte cell surface antigen-2 (ACSA-2) was shown to be highly expressed in this region [24]. To validate the specificity of our isolation protocol, in our RNAseq data we looked for the expression levels of genes associated with different CNS cell types and with different types of immune cells (Appendix A). We confirmed that this sorting method provided a cell population enriched in astrocytes, since the majority of astrocyte and Bergmann-glia-associated genes were highly expressed in our samples, while the majority of genes associated with other cell types were not expressed or presented low expression values. The ATPase Na+/K+ Transporting Subunit Beta 2 (Atp1b2) gene, which was identified as being the ACSA-2 epitope [25], also presented high expression levels.

PTX has been suggested to facilitate EAE development by increasing the blood brain barrier (BBB) permeability, thereby facilitating the migration of pathogenic T cells to the CNS, among other important biological effects [26,27]. Considering that astrocytes play an important role in the maintenance of BBB permeability [28,29], and that non-induced animals were also injected with CFA and PTX, we also sacrificed a group of non-induced animals at each of the experimental time points. In fact, we observed that CFA and PTX alone can have a role in the astrocytic response because several astrocytic genes were significantly altered in non-induced animals, when comparing the animals of the three experimental time points (Appendix A). Indeed, one disadvantage of the EAE model arises from the use of CFA and PTX for active disease induction. CFA contains bacterial components that are able to activate the innate immune system, via pattern recognition receptors [12], and, consequently, misrepresent the animals’ general immune reactivity and confound the findings related with regulatory mechanisms [30]. In the case of PTX, it will contribute to BBB permeabilization and facilitate autoantigen recognition in the CNS, by activating tissue-resident APCs [31].

Next, we compared gene expression values between EAE and non-induced animals, for each time point analyzed, and observed a total of 804 genes significantly altered at the pre-symptomatic phase, 470 genes altered at the onset phase, and 457 genes altered at the chronic phase of disease (Figure 1B). To explore the alterations occurring along with disease development, we also compared the three EAE time points among each other without normalizing first for the respective non-induced group that was sacrificed in each time point at the same time of the EAE sacrificed animals (Appendix A). However, some of the differences found were also present in the comparison between time points in non-induced animals. For that reason, we next normalized each disease time point for the respective non-induced group and verified which differences remained significant when comparing the EAE groups (Figure 1C). Of interest, the chronic and pre-symptomatic phases of disease were more similar among each other than with the onset phase.

Over-representation pathway analysis was performed for the differentially expressed genes between EAE and non-induced animals for each time point (Figure 2A–C, Appendix A) and between EAE time points (Figure 2D–F, Appendix A). For this pathway analysis, all DEGs were included independently of being specific for that time point or being shared among time points: (EAE vs. non-induced pre-symptomatic n = 804; EAE vs. non-induced onset n = 470 and EAE vs. non-induced chronic n = 457 and in the comparisons among the EAE time points: onset EAE vs. pre-symptomatic EAE n = 570; chronic EAE vs. pre-symptomatic EAE n = 132 and chronic EAE vs. onset EAE n = 296). Interestingly, several pathways related with extracellular matrix organization were significantly altered in EAE animals at the pre-symptomatic phase of disease, while at the onset phase, the mostly represented pathways were involved in metabolic pathways.

### 3.2. Astrocytic Metabolic Reprogramming at the Onset Phase of Disease

As already mentioned, several metabolic pathways were significantly altered at the onset phase of disease. Among these were genes involved in glycolysis and the tricarboxylic acid cycle (TCA) cycle (Figure 3A). Using a different group of animals, we confirmed by qRT-PCR that aldolase, fructose-bisphosphate C (*Aldoc*), and isocitrate dehydrogenase (NAD+) 3 non-catalytic subunit gamma (*Idh3g*) were overexpressed in EAE animals at the onset time point, while we observed a tendency for increased expression of phosphofructokinase (*Pfkm*) and succinate dehydrogenase complex flavoprotein subunit A (*Sdha*) at this time point (Figure 3B, Table 1. Two-Way ANOVA—between subject factors: disease and time point).

Recently, Liddelow and colleagues (2017) characterized a population of reactive neurotoxic astrocytes (A1), induced by classically activated neuroinflammatory microglia [19]. In our samples, at the onset phase, we observed overexpression only of astrocytic genes such as *Fibulin 5* (*Fbln5*), *adhesion molecule with Ig like domain 2* (*Amigo2*) and *FK506 binding protein 5* (*Fkbp5*) which were previously shown, among others, to be related with astrocytic neurotoxic reactivity [19] (Figure 4A,B, Table 1. Two-Way ANOVA—between subject factors: disease and time point).

Classically activated (M1) macrophages are characterized by an induction of aerobic glycolysis, which results in lactate production and increased levels of TCA cycle intermediates [32]. Considering that similar metabolic alterations occur in astrocytes of EAE animals, and that these cells acquire a neurotoxic phenotype, it is possible that astrocytes could also undergo metabolic reprogramming in the EAE context.

### 3.3. Increased Astrocytic Length and Complexity in EAE Animals near Lesion Regions

In EAE animals, at the onset and chronic phases, it was possible to observe focal perivascular lesions in the cerebellum white matter, absent in non-induced animals and at the pre-symptomatic phase of EAE. However, at those disease time points, the animals also presented regions where the white matter appeared normal (NAWM regions). To explore the morphological alterations occurring in astrocytes after disease induction, we reconstructed their morphology in the cerebellum white matter of EAE animals, around lesion regions and also in NAWM regions, and compared them with astrocytes from non-induced animals. Figure 5A shows representative images of the GFAP staining in non-induced animals, EAE animals sacrificed at the pre-symptomatic phase, and NAWM and lesion regions of EAE animals sacrificed at the onset and chronic phases. The stacked images were used to perform a three-dimensional reconstruction of astrocytes, and the representative drawings for each experimental group are shown in Figure 5B.

We started by comparing astrocytes from the NAWM of EAE animals with non-induced animals and no significant differences were observed in the total astrocytic length (Figure 5C, Table 2. Two-Way ANOVA—between subject factors: disease and time point.). In addition, there were no differences in astrocytic ramification, as evaluated by the Sholl analysis, between the NAWM region of EAE animals and non-induced mice (Figure 5F, Table 3. Mixed ANOVA—between subject factors: disease, time point; within subject factor: radius).

In lesioned areas of the EAE cerebellum, astrocytes were longer (Figure 5D, Table 2 Two-Way ANOVA—between subject factors: disease and time point) and more ramified compared to non-induced controls (Figure 5G, Table 3. Mixed ANOVA—between subject factors: disease, time point; within subject factor: radius).

We also observed an overall tendency for increased astrocytic length (Figure 5E, Table 2. Two-Way repeated measures ANOVA—between subject factor: time point; within subject factor: white matter region) and ramification near lesion regions compared to NAWM regions (Figure 5H, Table 3. Mixed ANOVA—between subject factor: time point; within subject factors: white matter region, radius).

## 4. Discussion

By performing high-throughput sequencing of astrocytes’ transcriptomes during EAE disease progression, we show that astrocytes isolated from the cerebellum displayed a strong alteration in the genes that encode for proteins related to several metabolic pathways. Specifically, we found a significant increase in glycogen degradation, glycolytic, and TCA cycle enzymes. Together with these alterations, we detected an upregulation in genes that characterize “astrocyte reactivity”. How these metabolic alterations are linked with a pro-inflammatory and neurotoxic astrocytic phenotype and disease progression is unknown, but the metabolic alterations that we documented here and their possible correlation with astrocytic phenotype has been known for a while in immune cells, especially monocytes—a field termed immunometabolism. In this field, it was shown that a monocytic increase in TCA cycle is associated with a stronger immune response and a monocytic decrease in TCA cycle is associated with immune tolerance. Indeed, metabolic reprogramming is known to occur in macrophages and microglia in response to different stimuli. M1 (classically activated) macrophages/microglia are activated by bacterial-derived products, like LPS, and infection-associated signals, such as IFN gamma, and are usually part of the first line of defense of the innate immune system [32,33]. M1 polarized cells then produce pro-inflammatory cytokines and high levels of NO in order to kill the foreign pathogen and activate T cells to mount an adaptive immune response. A similar response can occur in the absence of microorganisms, as a result of trauma, ischemia-reperfusion injury, or chemical exposure [33]. On the other hand, M2 (alternatively activated) macrophages play an important role in the resolution phase of inflammation [32,33], by producing anti-inflammatory factors that switch off pro-inflammatory cell phenotypes and re-establish homeostasis [33]. Notably, the cells’ metabolic profile is a reflection of these functions. Namely, in M1 cells, aerobic glycolysis is induced, to provide the cell with rapid energy, and the pentose phosphate pathway and respiratory chain functions are induced and attenuated, respectively, to increase the production of ROS and RNS [21,22]. M2 polarized cells need to be sustained for longer periods of time, so oxidative metabolism and fatty acid oxidation are induced [32,33]. Interestingly, the polarization for one of these phenotypes can be induced not only by the inflammatory factors already mentioned, but also by the modification of the cell’s metabolic state. Specifically, blocking oxidative metabolism drives macrophage polarization to an M1 state, while forcing it in M1 macrophages potentiates the M2 phenotype [32].

Of relevance, astrocytes were recently classified into “A1” (neurotoxic) and “A2” (neuroprotective), in analogy to the M1/M2 macrophage nomenclature [19]. Liddelow and colleagues (2017) showed that neuroinflammation induced an A1 phenotype, characterized by the loss of normal astrocytic functions and the gain of a neurotoxic role, which was deleterious for both neurons and oligodendrocytes. Conversely, A2 astrocytes were induced by ischemic conditions and presented a neuroprotective phenotype [19]. Additionally, C3 + A1 astrocytes were found to be present in demyelinating lesions of MS patients [19]. In accordance, we observed that astrocytes isolated from the cerebellum of EAE animals presented an A1 phenotype, particularly at the onset phase of disease. Moreover, several genes involved in metabolic pathways were altered during this time point, including glycolytic and TCA cycle genes. Previous studies had also reported alterations in other metabolic pathways, namely sphingolipid metabolism and cholesterol biosynthesis, in astrocytes during EAE, which we also observed here [34,35,36,37,38,39]. Considering all these data, and the fact that metabolic alterations are associated with M1/M2 polarization, we hypothesize that similar metabolic alterations could occur in A1/A2 astrocytes.

Of interest, in vitro treatment of astrocytes with dimethyl itaconate (known to modulates TCA and redox metabolism) alters the astrocyte phenotype from A1 neurotoxic to A2 neuroprotective [40]. In addition, a recent in vitro study has demonstrated differential metabolic profiles in astrocytes exposed to LPS for short or long periods of time. Namely, 30 min of LPS treatment increased the astrocytic glycolytic rate but did not affect their oxidative phosphorylation rate, while treatment with LPS for 24 h decreased the glycolytic capacity and increased the mitochondrial respiration of astrocytes [41]. This study suggests that pro-inflammatory stimuli are able to modulate the metabolic profile of astrocytes, supporting the findings observed in our work.

In this work, we also performed for the first time a morphological analysis of astrocytes of the cerebellum white matter in different disease time points. As already expected, astrocytes near lesion regions, at the onset and chronic phases of disease, were more reactive than astrocytes from non-induced animals, as evaluated by the increased total length and number of ramifications. Regarding the NAWM, previous studies have reported the presence of gliosis in MS patients [42]; however, Graumann and colleagues (2003) did not observe significant differences in GFAP expression between the NAWM of MS patients and control subjects [43]. In accordance, we also did not find significant differences in the astrocytic morphology between the NAWM of EAE animals and non-induced controls.

Altogether, we propose that, in response to EAE, astrocytes showed an increased expression of enzymes of glycolysis and the TCA cycle suggesting a metabolic shift to oxidative phosphorylation. These alterations are concomitant with the development of a neurotoxic phenotype. Hence, we hypothesize that TCA cycle upregulation in MS astrocytes contributes to exacerbating brain inflammation, leading to myelin damage. Thus, a new disease mechanism tackling glia metabolic reprograming is hypothesized and needs to be addressed in the future to halt disease progression.

## 5. Conclusions

In this research, we used an MS animal model to perform high-throughput sequencing of astrocytes’ transcriptome during disease progression. Our data show a surprising upregulation of glycolytic and TCA cycle regulatory genes. Together with these alterations, we detected an upregulation of genes that characterize “astrocyte reactivity”. These results are in line with data from other fields, showing that a TCA cycle increase in monocytes is associated with a stronger immune response. Hence, herein we raised a new possible hypothesis in the MS field in which TCA cycle upregulation in MS astrocytes may contribute to exacerbating brain inflammation leading to myelin damage. This hypothesis still needs further confirmation with other MS models and further validations.

## Figures and Tables

**Figure 1 cells-12-02484-f001:**
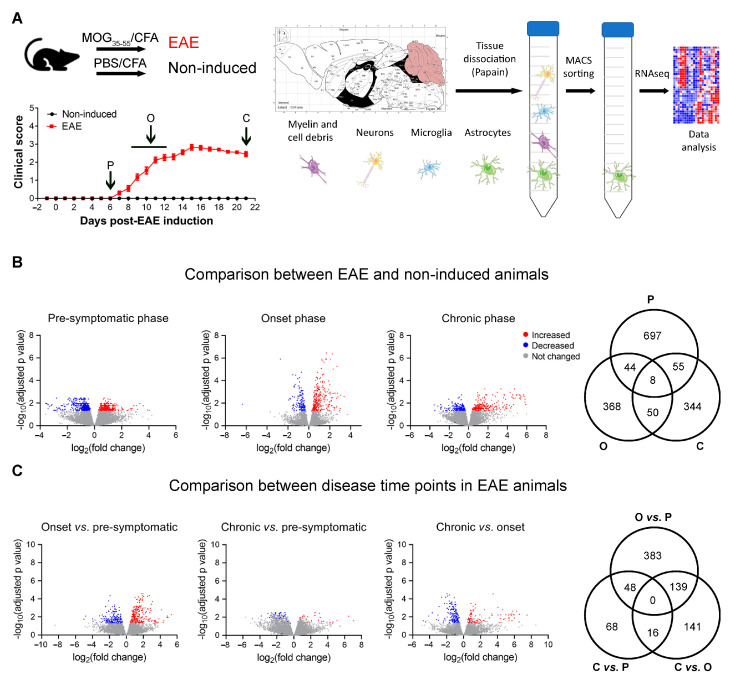
Sample collection for RNAseq analysis and differentially expressed genes. (**A**) Schematic overview of the methodology used to isolate astrocytes from the cerebellum and posterior RNAseq analysis; average clinical score of the animals used in the study. (**B**) Volcano plot and Venn diagram depicting the differentially expressed astrocytic genes after comparing EAE and non-induced animals for each experimental time point. (**C**) Volcano plot and Venn diagram depicting the differentially expressed genes after comparison between disease time points in EAE animals, normalized for the respective non-induced group. In the volcano plots, significant downregulated genes are represented in blue; significant upregulated genes are represented in red; genes whose expression was not significantly altered are represented in gray. Images obtained and adapted from Servier Medical Art. CFA—complete Freund’s adjuvant; C—chronic time point; MACS—magnetic-activated cell sorting; MOG—myelin oligodendrocyte glycoprotein; O—onset time point; PBS—phosphate buffered saline; P—pre-symptomatic time point.

**Figure 2 cells-12-02484-f002:**
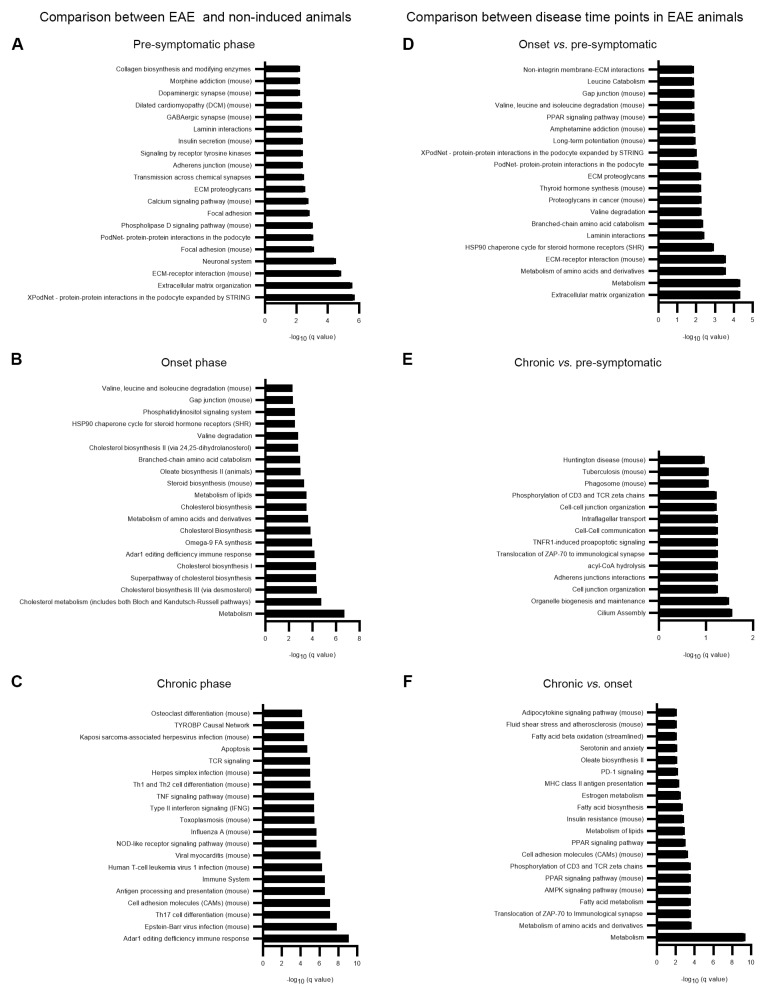
Pathway analysis results. (**A**) Top 20 pathways altered in EAE animals at the pre-symptomatic, (**B**) onset, and (**C**) chronic phases of disease, compared to non-induced animals. (**D**) Top 20 pathways altered at the onset vs. pre-symptomatic phases. (**E**) Top 20 pathways altered at the chronic vs. pre-symptomatic phases. (**F**) Top 20 pathways altered at the chronic vs. onset phases.

**Figure 3 cells-12-02484-f003:**
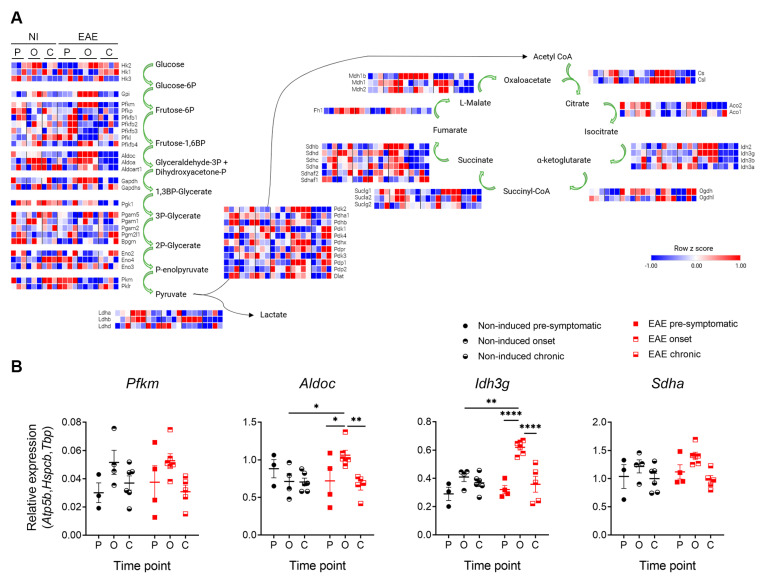
Increased expression of metabolic genes in astrocytes from EAE animals. (**A**) Representation of the enzymes involved in glycolysis and the TCA cycle and respective RNAseq expression levels. (**B**) qRT-PCR expression levels of enzymes involved in glycolysis and the TCA cycle, namely Pfkm, Aldoc, Idh3g, and Sdha. Data are presented as the mean ± SEM. n = 3–4 in (**A**) and n = 3–6 in (**B**). * *p* < 0.05, ** *p* < 0.01, **** *p* < 0.0001. Heatmaps obtained using the Morpheus APP tool of CLUE, Broad Institute, Cambridge, MA, USA. C—chronic time point; NI—non-induced; O—onset time point; P—pre-symptomatic time point.

**Figure 4 cells-12-02484-f004:**
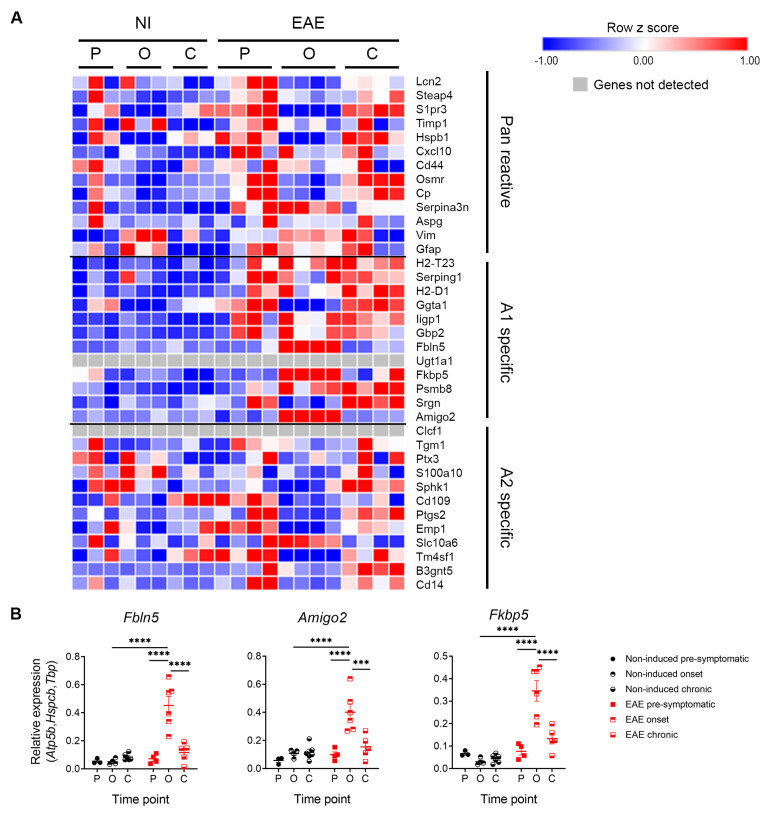
Astrocytes from EAE animals presented a neurotoxic phenotype. (**A**) RNAseq expression levels of Pan reactive, neurotoxic (A1) specific and neuroprotective (A2) specific genes. (**B**) qRT-PCR expression levels of A1-specific genes. Data are presented as the mean ± SEM. n = 3–4 in (**A**) and n = 3–6 in (**B**). *** *p* < 0.001, **** *p* < 0.0001. Heatmaps obtained using the Morpheus APP tool of CLUE. C—chronic time point; NI—non-induced; O—onset time point; P—pre-symptomatic time point.

**Figure 5 cells-12-02484-f005:**
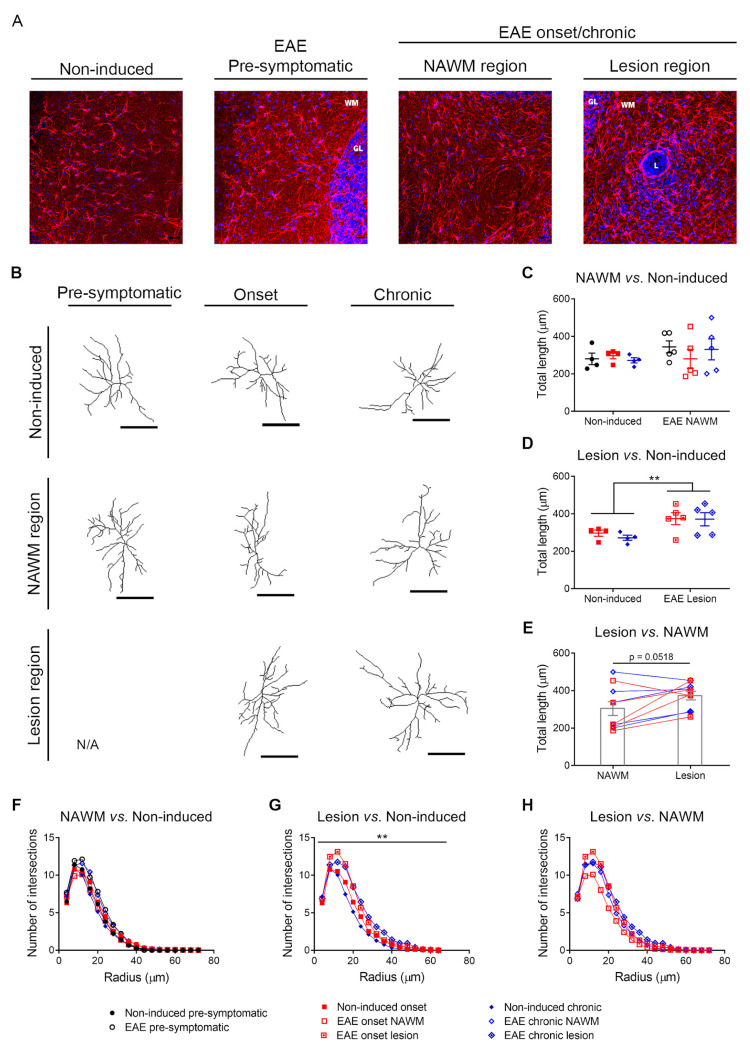
Astrocytes near lesion regions, in EAE animals, were longer and more complex compared to astrocytes from non-induced animals. (**A**) Representative images of cerebellum sections from non-induced and EAE animals immunostained for GFAP (scale bar represents 20 µm). (**B**) Representative drawings of astrocytes reconstructed using the Simple Neurite Tracer plugin of Fiji (scale bar represents 20 µm). (**C**) Comparison between the astrocytic total process length of non-induced animals and the NAWM regions of EAE animals. (**D**) Comparison between the astrocytic total process length of non-induced animals and the lesion regions of EAE animals. (**E**) Comparison between the total astrocytic process length of NAWM and the lesion regions of EAE animals. (**F**) Comparison between the number of intersections per radius in astrocytes from non-induced animals and from the NAWM regions of EAE animals. (**G**) Comparison between the number of intersections per radius in astrocytes from non-induced animals and from the lesion regions of EAE animals. (**H**) Comparison between the number of intersections per radius in astrocytes from the NAWM and lesion regions of EAE animals. Data are presented as the mean ± SEM for total length or as the mean for Sholl analysis. n = 4–5, average of 7–8 astrocytes per animal. ** *p* < 0.01 for comparison between lesion region in EAE animals (onset and chronic phases) and non-induced groups (onset and chronic phases). GL—granular layer; L—lesion; N/A—not applicable; NAWM—normal appearing white matter; WM—white matter.

**Table 1 cells-12-02484-t001:** Statistical analysis for the qRT-PCR data.

	Disease			Time Point			Interaction		
	F Value	*p* Value	Effect Size	F Value	*p* Value	Effect Size	F Value	*p* Value	Effect Size
*Pfkm*	F_(1,22)_ = 0.0212	0.8854		F_(2,22)_ = 4.858	0.0179	η^2^ = 0.297	F_(2,22)_ = 0.4698	0.6312	
*Aldoc*	F_(1,22)_ = 0.3990	0.5341		F_(2,22)_ = 2.832	0.0805		F_(2,22)_ = 4.161	0.0293	η^2^ = 0.229
*Idh3g*	F_(1,22)_ = 6.639	0.0172	η^2^ = 0.089	F_(2,22)_ = 17.140	<0.0001	η^2^ = 0.462	F_(2,22)_ = 5.664	0.0104	η^2^ = 0.153
*Sdha*	F_(1,22)_ = 0.9211	0.3476		F_(2,22)_ = 5.358	0.0127	η^2^ = 0.309	F_(2,22)_ = 0.5214	0.6009	
*Fbln5*	F_(1,22)_ = 21.370	0.0001	η^2^ = 0.214	F_(2,22)_ = 12.670	0.0002	η^2^ = 0.253	F_(2,22)_ = 15.670	<0.0001	η^2^ = 0.313
*Amigo2*	F_(1,22)_ = 15.070	0.0008	η^2^ = 0.209	F_(2,22)_ = 10.150	0.0008	η^2^ = 0.281	F_(2,22)_ = 7.406	0.0035	η^2^ = 0.205
*Fkbp5*	F_(1,22)_ = 34.860	<0.0001	η^2^ = 0.326	F_(2,22)_ = 9.893	0.0009	η^2^ = 0.185	F_(2,22)_ = 15.170	<0.0001	η^2^ = 0.284

Legend: *Aldoc*—fructose-biphosphate aldolase C; *Amigo2*—adhesion molecule with Ig like domain 2; *Fbln5*—fibulin 5; *Fkbp5*—FK506 binding protein 5; *Idh3g*—isocitrate dehydrogenase 3 (NAD+) gamma; *Pfkm*—phosphofructokinase muscle; *Sdha*—succinate dehydrogenase complex subunit A flavoprotein. Italics are refering to the gene.

**Table 2 cells-12-02484-t002:** Statistical analysis for astrocytic total length data when comparing astrocytes from the NAWM of EAE animals with non-induced animals.

	Disease			Time Point			Interaction		
	F Value	*p* Value	Effect Size	F Value	*p* Value	Effect Size	F Value	*p* Value	Effect Size
Total length NAWM in EAE vs. non-induced	F_(1,21)_ = 1.158	0.2941		F_(2,21)_ = 0.174	0.8415		F_(2,21)_ = 0.637	0.5389	
Total length lesion in EAE vs. non-induced	F_(1,14)_ = 9.548	0.0080	η_p_^2^ = 0.405	F_(1,14)_ = 0.243	0.6297		F_(1,14)_ = 0.149	0.7050	
	**White matter region**			**Time point**			**Interaction**		
Total length lesion in EAE vs. NAWM in EAE	F_(1,8)_ = 5.214	0.0518	η_p_^2^ = 0.395	F_(1,8)_ = 0.182	0.6809		F_(1,8)_ = 0.821	0.3915	

**Table 3 cells-12-02484-t003:** Statistical analysis for Sholl analysis of astrocytes when comparing the NAWM region of EAE animals and non-induced mice.

					Interaction			
		F Value	*p* Value	Effect Size		F Value	*p* Value	Effect Size
NAWM in EAE vs. non-induced	Disease	F_(1,21)_ = 0.939	0.3435		radius*disease	F_(17,357)_ = 1.049	0.4034	
	Time point	F_(2,21)_ = 0.159	0.8544		radius* time point	F_(34,357)_ = 0.479	0.9946	
	Radius	F_(17,357)_ = 354.805	<0.0001	η_p_^2^ = 0.944	radius*disease* time point	F_(34,357)_ = 0.793	0.7920	
					disease* time point	F_(2,21)_ = 0.762	0.4794	
Lesion in EAE vs. non-induced	Disease	F_(1,14)_ = 9.201	0.0089		radius*disease	F_(15,210)_ = 4.685	<0.0001	η_p_^2^ = 0.251
	Time point	η_p_^2^ = 0.397	F_(1,14)_ = 0.104	0.7524	radius* time point	F_(15,210)_ = 0.903	0.5613	
	Radius	F_(15,210)_ = 330.458	<0.0001	η_p_^2^ = 0.959	radius*disease* time point	F_(15,210)_ = 1.061	0.3949	
					disease* time point	F_(1,14)_ = 0.490	0.4952	
Lesion in EAE vs. non-induced	White matter region	F_(1,8)_ = 4.418	0.0687		radius*region	F_(17,136)_ = 2.991	0.0002	η_p_^2^ = 0.272
	Time point	F_(1,8)_ = 0.426	0.5321		radius* time point	F_(17,136)_ = 0.299	0.9969	
	Radius	F_(17,136)_ = 153.157	<0.0001	η_p_^2^ = 0.950	radius*region* time point	F_(17,136)_ = 1.673	0.0550	
					region* time point	F_(1,8)_ = 0.548	0.4802	

* represents interaction between the factors mentioned.

## Data Availability

The data presented in this study are available in within the article or Appendix A.

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
