# Peer review of "Astrocytes Undergo Metabolic Reprogramming in the Multiple Sclerosis Animal Model"

_cells, 2023, doi:10.3390/cells12202484_

Round 1

Reviewer 1 Report

In this work, the authors showed alterations in genes encoding metabolic proteins in astrocytes derived from cerebellar tissue of mice affected by a model of EAE. These changes may reflect a particular alteration in the phenotype of these astrocytes and may be related to the changes seen in patients with multiple sclerosis. It is an interesting work with interesting results. However, it is necessary to clarify some aspects.

1.  Introduction – The first two paragraphs refer only multiple sclerosis in humans, whereas this is a paper on an experimental animal and in vitro model. Therefore, the information in lines 33-48 should be shortened and some general information on cerebellar astrocytes and their involvement in MS should be included.

2.  Introduction – In lines 51-52, the authors argue that “there is no peripheral immune cells invasion into the CNS”. However, further down (in lines 200-202, they argue that “to facilitate EAE development by increasing the BBB permeability, thereby facilitating the migration of pathogenic T cells to the CNS”… So how can it make sense or be useful to test a hypothesis with a model in which the opposite happens? Please explain or resolve this apparent controversy.

3.  Results – Although understandable later, the mention of ACSA-2 in lines 190-191 is confusing. Please rephrase.

4.  Lines 205-206 So what? The sentence is incomplete. It needs a little more explanation of the possible meaning.

5.  Line 210. the mention of 296 seems wrong. Aren’t we 457 genes instead?

6.  Why does normalizing the data seem to have a very different effect at different times in the study? Please elaborate on possible explanations.

7.  What might be the biological significance of the eight genes in Figure 1 that vary together in the three groups studied?

8.  Line 232. how many genes were included in the pathway analysis? Please indicate for each case or comparison.

9.  Figure 1 panel B. Were these data normalized? How? (As mentioned for Figure 1 panel C.

10.         Line 246. “The overexpression of some of these genes”… How many? It seems that only two genes were confirmed and two others were not.

11.         Lines 249 to 258 It would be better to present this in a table than in a paragraph.

12.         Lines 269-270. in the sentence “In our samples, we observed overexpression of genes specific for this neurotoxic phenotype in EAE animals." ... How can the author assume this? This needs to be described and explained in more detail.

13.         Line 271. on what basis do the authors refer to the genes as “neuroprotective (A2)-specific genes”?

14.         Lines 321-354. All statistical comparisons would be better listed in a table rather than in the text (information in lines: 328-332; 337-342; 344-353).

none

Author Response

Reviewer: 1

Comments to the Author

In this work, the authors showed alterations in genes encoding metabolic proteins in astrocytes derived from cerebellar tissue of mice affected by a model of EAE. These changes may reflect a particular alteration in the phenotype of these astrocytes and may be related to the changes seen in patients with multiple sclerosis. It is an interesting work with interesting results. However, it is necessary to clarify some aspects.

We acknowledge the reviewer remarks, comments and suggestions that greatly contribute to improve the manuscript.

Major:
1.
Introduction – The first two paragraphs refer only multiple sclerosis in humans, whereas this is a paper on an experimental animal and in vitro model. Therefore, the information in lines 33-48 should be shortened and some general information on cerebellar astrocytes and their involvement in MS should be included.

We acknowledge the reviewer comment and now we revised the introduction in accordance. We shortened the information regarding the human data and we add in line 69-73 information on the relevance of cerebellar astrocytes in MS/EAE: “The study of cerebellar astrocytes in the MS is relevant since cerebellar abnormalities are continuously demonstrated to be associated with a variety of motor or non-motor dysfunctions. Also, we previously showed that, specifically, cerebellar astrocytes are activated in the experimental autoimmune encephalomyelitis (EAE) MS animal model and that their activation is associated with the EAE motor problems [7].”.

  1. Introduction – In lines 51-52, the authors argue that “there is no peripheral immune cells invasion into the CNS”. However, further down (in lines 200-202, they argue that “to facilitate EAE development by increasing the BBB permeability, thereby facilitating the migration of pathogenic T cells to the CNS”… So how can it make sense or be useful to test a hypothesis with a model in which the opposite happens? Please explain or resolve this apparent controversy.

We acknowledge the fact that the reviewer highlighted this apparent controversy and we agree with the reviewer. Indeed, this is related with the limitations when using animal models to study human diseases. However, to avoid this point, elegantly raised by the reviewer, we rephrase the introduction. Now, in the revised manuscript, we rephrase 55-59: “The reason for this unsatisfactory situation is that the disease mechanisms driving progressive MS remain unresolved. However, one intriguing observation is the fact that, in patients in the progressive stage of the disease, local glial cells are also relevant and able to potentiate inflammation alone, leading to neurodegeneration by themselves which suggests that glial cells are key in MS [3].”

  1. Results – Although understandable later, the mention of ACSA-2 in lines 190-191 is confusing. Please rephrase.

We acknowledge the correction and in the revised manuscript in line 222 we specified “The cerebellum is known to be affected both in the human disease and in the EAE model, and astrocyte cell surface antigen-2 (ACSA-2) was shown to be highly expressed in this region [24].”

  1. Lines 205-206 So what? The sentence is incomplete. It needs a little more explanation of the possible meaning.

We now clarified that in the revised manuscript. Line 238 of the revised manuscript this clarification was added: “In fact, we observed that CFA and PTX alone can have a role in the astrocytic response because several astrocytic genes were significantly altered in non-induced animals, when com-paring the animals of the three experimental time points (Supplementary file 2 – Supplementary figure 2A).”

  1. Line 210. the mention of 296 seems wrong. Aren’t we 457 genes instead?

We really acknowledge the correction and in the revised manuscript we corrected the value to 457 genes (line 250).

  1. Why does normalizing the data seem to have a very different effect at different times in the study? Please elaborate on possible explanations.

In this study we did three normalizations. The first normalization was performed when we compared the EAE group with the non-induced animals. This comparison was possible because in each time point that we sacrificed EAE animals we also had a group of non-induced animals that were sacrificed at the same time point (at pre-onset, onset and chronic phase). This comparison gave us a list of genes that were altered for each time point of the disease and that was due to the EAE induction.

The other normalization was different because we only took in consideration the list of genes that were altered in each time point of the EAE animals, after normalization to the respective non-induced animals, and we did comparisons among the different phases of the disease. In the data that we present in supplementary figure 2B we did this same comparison (among the different phases of the disease) but using the list of genes altered in each time point but without normalization for the non-induced animals. We now clarified this point in the revised manuscript line 252-254.

  1. What might be the biological significance of the eight genes in Figure 1 that vary together in the three groups studied?

We understand the point raised by the reviewer. The eight genes that were commonly altered among the three groups were: Xdh, Stat1, Cckbr, Kif5a, Dmp1, Acot1, F2r, Fsd1l. With this very low number of altered genes were not even possible to perform a pathways analysis which may suggest that their biological relevance is probably very limited. Also, because of that we only showed in the figure 1 the number and we didn’t highlight even the genes because of that reason.

  1. Line 232. how many genes were included in the pathway analysis? Please indicate for each case or comparison.

All DEGs were included in pathway analysis, independently of being specific for that time point or being shared among time points:

  • EAE vs NI pre-symptomatic n= 804 DEGs
  • EAE vs NI onset n = 470
  • EAE vs NI chronic n = 457
  • Onset EAE vs pre-symptomatic EAE n = 570
  • Chronic EAE vs pre-symptomatic EAE n = 132
  • Chronic EAE vs onset EAE n = 296

We add that information in the revised manuscript (line 283-287).

  1. Figure 1 panel B. Were these data normalized? How? (As mentioned for Figure 1 panel C.

Data in panel B was not normalized. In B we wanted to study what genes are altered in EAE animals compared to non-induced controls. In C, we wanted to compare the different EAE time points, but we also found alterations between the 3 time points among the non-induced groups. So, some alterations observed in the comparisons between the EAE animals could be associated with basal alterations, and not be dependent on the diseased condition. Thus, we first normalized each EAE group for the respective non-induced time point, and only after performed comparisons between time points. We now clarified this point in the revised manuscript line 252-254.

  1. Line 246. “The overexpression of some of these genes”… How many? It seems that only two genes were confirmed and two others were not.

We now rephrase this sentence in the revised manuscript, line 299-304: “Using a different group of animals, we confirmed by qRT-PCR that aldolase, fructose-bisphosphate C (Aldoc) and isocitrate dehydrogenase (NAD+) 3 non-catalytic subunit gamma (Idh3g) were overexpressed in EAE animals at the onset time point, while we observed a tendency for increased expression of phosphofructokinase (Pfkm) and succinate dehydrogenase complex flavoprotein subunit A (Sdha) at this time point”.

  1. Lines 249 to 258 It would be better to present this in a table than in a paragraph.

We acknowledge the reviewer suggestion and in the revised manuscript we provided a new Table 1.

  1. Lines 269-270. in the sentence “In our samples, we observed overexpression of genes specific for this neurotoxic phenotype in EAE animals." ... How can the author assume this? This needs to be described and explained in more detail.

We acknowledge the point raised by the reviewer and we now tried to explain it more clearly. We rephrase that sentence in line 316-320:  “In our samples, at the onset phase, we observed overexpression only of astrocytic genes such as Fibulin 5 (Fbln5), adhesion molecule with Ig like domain 2 (Amigo2) and FK506 binding protein 5 (Fkbp5) which were previously shown, among others, to be related with astrocytic neurotoxic reactivity [17].”.

  1. Line 271. on what basis do the authors refer to the genes as “neuroprotective (A2)-specific genes”?

To answer to the reviewer last point we rephrase and simplify that sentence and the reference to the neuroprotective (A2)-specific genes were now deleted. But as it is stated thought the text this classification was proposed by Liddelow et al. 2017 (Nature 2017, 541, 481-487, doi:10.1038/nature21029.)

  1. Lines 321-354. All statistical comparisons would be better listed in a table rather than in the text (information in lines: 328-332; 337-342; 344-353).

We acknowledge the reviewer suggestion and in the revised manuscript we provided a new Table 2 and Table 3.

Reviewer 2 Report

The study researchers must specify the aspects related to the use of the Experimental Autoimmune Encephalomyelitis (EAE) induction method. They must mention bibliographic references because the methodology is known and specific aspects for the changes determined in experimental animals and they need to justify the choice only of female mice.

I consider the topic relevant in the field and they tried to find changes specific to the pathology, but many further experiments are needed to establish a causal link.

Another difficult aspect to follow in the experiment is the number of animals for the six groups used and the evaluated parameters.

In line 316 - "Data presented as mean ± SEM for total length or as mean for sholl analysis n = 4-5, average of 7-8 astrocytes per animal." Why are there only n=4 -5? The number of animals per group is not clearly specified to see exactly.For statistical significance, a number of 4-5 is small when the SD is large.

The conclusions are based on the results shown and are in accordance with the working hypothesis. But the researchers must present the given data more clearly by organizing the results in tables in which the parameters for the working groups can be more clearly observed.

It is necessary to revise these aspects so that the article can be published.

Author Response

Reviewer: 2

The study researchers must specify the aspects related to the use of the Experimental Autoimmune Encephalomyelitis (EAE) induction method. They must mention bibliographic references because the methodology is known and specific aspects for the changes determined in experimental animals and they need to justify the choice only of female mice.

We acknowledge the reviewer remarks, comments and suggestions that contribute to improve the manuscript.

Regarding the EAE animal model, in the revised manuscript we now add information to better clarify the animal model and also their limitations.

We add that information in the materials and methods section line 90-100: “As previously shown, C57BL/6 mice induced with MOG35-55 present a chronic course of disease [8,9] characterized by ascending paralysis resultant from the preferential attack to the spinal cord [10]. The disease is characterized in the beginning by a limp tail, which progresses to hind and forelimbs paralysis [11]. Also, multifocal and confluent areas of mononuclear inflammatory infiltrates and perivascular inflammatory cuffing in the cerebellum and hindbrain white matter are observed. The chronic disease course makes this a good model to study SP-MS [12]. The MOG35-55-induced C57BL/6 model is particularly important in MS studies due to the increased availability of gene- modified strains on this background [13,14]. Herein, we used females for the immunization, because females have more spinal cord infiltrating cells and demyelination than males in spite of essentially identical EAE disease severity [15].”

But also in the results section line 241-247: “Indeed, one disadvantage of the EAE model arises from the use of CFA and PTX for active disease induction. CFA contains bacterial components that are able to activate the innate immune system, via pattern recognition receptors [12], and, consequently, misrepresent the animals’ general immune reactivity and confound the findings related with regulatory mechanisms [30]. In the case of PTX, it will contribute to BBB permeabilization and facilitate autoantigen recognition in the CNS, by activating tissue-resident APCs [31].”.

I consider the topic relevant in the field and they tried to find changes specific to the pathology, but many further experiments are needed to establish a causal link.

We agree with the reviewer comment and, in the revised manuscript, we highlighted that limitation of this study. We, now, highlighted that our results only raised new possible hypothesis for the disease modulation. We now rephrase the conclusion section: “Hence, herein we raised a new possible hypothesis in the MS field in which, the TCA cycle upregulation in MS astrocytes, may contribute to exacerbate brain inflammation leading to myelin damage. This hypothesis still needs further confirmation with other MS models and further validations.”

Another difficult aspect to follow in the experiment is the number of animals for the six groups used and the evaluated parameters.

We acknowledge the suggestion and the number of animals is, now, specified in more detail in the Methods section. In line 153 we have previously stated that: “A total of three non-induced and four EAE animals were sacrificed per experimental time point for the RNAseq experiments.” But, in the revised manuscript we now add in line 178-181 the following: “Primers DNA sequences and annealing temperatures are provided in Supplementary file 2 – Supplementary table 1. A total of three animals were used in the non-induced pre-symptomatic group; four animals in the non-induced onset and EAE pre-symptomatic groups; five animals in the EAE chronic group; and six animals in the non-induced chronic and EAE onset groups.”. In line 203-205 we also add: “Results are presented as the average of 4-5 animals per experimental group (n = 4 for non-induced groups and n = 5 for EAE groups), and 7-8 astrocytes were reconstructed per animal. The average of all astrocytes per animal was used for statistical analysis.”

In line 316 - "Data presented as mean ± SEM for total length or as mean for sholl analysis n = 4-5, average of 7-8 astrocytes per animal." Why are there only n=4 -5? The number of animals per group is not clearly specified to see exactly.For statistical significance, a number of 4-5 is small when the SD is large.

We have evaluated 7-8 astrocytes per animal, and have used the average value for each animal in statistical comparison between experimental groups. In the control non-induced groups the SD is much smaller than the SD in the EAE groups, which could just reflect a higher biological variability in the diseased animals, which is somehow typically and a limitation of this animal model.

The conclusions are based on the results shown and are in accordance with the working hypothesis. But the researchers must present the given data more clearly by organizing the results in tables in which the parameters for the working groups can be more clearly observed.

We acknowledge the reviewer suggestion and we now add new tables in the revised manuscript. A new Table 1, 2 and 3 is now provided.

Reviewer 3 Report

In their paper entitled “Astrocytes undergo metabolic reprogramming in the Multiple Sclerosis animal model”, the Authors report that, in an animal model of multiple sclerosis (MS), astrocytes isolated from the cerebellum show a strong alteration in genes encoding for proteins involved in metabolic pathways. In particular, they found a significant increase of glycogen degradation, and of glycolytic and TCA cycle enzymes, while, on the other hand, up-regulation of genes involved in astrocyte reactivity was noticed. Finally, the Authors also report that near lesions astrocytes presented increased length and complexity, compared to control astrocytes.

 The paper is of interest and suitable for Cells. Both Methods and Results are well described, and the figures are highly informative. I only have a very minor  suggestion: throughout the text, and also in the figure legends, the first time a molecule (for example ACSA-2), or a structure (for example, BBB) is cited, it is better to report its complete name instead of the acronym only.

Author Response

Reviewer: 3

In their paper entitled “Astrocytes undergo metabolic reprogramming in the Multiple Sclerosis animal model”, the Authors report that, in an animal model of multiple sclerosis (MS), astrocytes isolated from the cerebellum show a strong alteration in genes encoding for proteins involved in metabolic pathways. In particular, they found a significant increase of glycogen degradation, and of glycolytic and TCA cycle enzymes, while, on the other hand, up-regulation of genes involved in astrocyte reactivity was noticed. Finally, the Authors also report that near lesions astrocytes presented increased length and complexity, compared to control astrocytes.

 The paper is of interest and suitable for Cells. Both Methods and Results are well described, and the figures are highly informative. I only have a very minor  suggestion: throughout the text, and also in the figure legends, the first time a molecule (for example ACSA-2), or a structure (for example, BBB) is cited, it is better to report its complete name instead of the acronym only.

We acknowledge the reviewer comments and the corrections were included in the revised manuscript.

Round 2

Reviewer 1 Report

The authors have addressed most of the questions and comments, consequently the manuscript has been improved in precision.

None